

**Reshaped acclimation traits of dominant tree species under manipulated rainfall**
**would alter their coexisting relation in a low-subtropical secondary evergreen**
**forest**
Lei Ouyang, Jianguo Gao, Ping Zhao*, Weijun Shen, Xingquan Rao, Liwei Zhu,
Guangyan Ni
Key Laboratory of Vegetation Restoration and Management of Degraded Ecosystems,
South China Botanical Garden, Chinese Academy of Sciences, Guangzhou 510650,
China
Center of Plant Ecology, Core Botanical Gardens, Chinese Academy of Sciences,
Guangzhou 510650, China
*Corresponding author.    Tel.: +86 20 37252881; fax: +86 20 37252831.
E-mail address: zhaoping@@scib.ac.cn



**Abstract.** This study explores the seasonal transpiration and physiological responses
of two dominant coexisting tree species, *Schima superba* and *Michelia macclurei*, to
manipulated precipitation patterns in a subtropical evergreen broadleaf forest of South
China, in which an ambient control treatment (BC), a drier dry and wetter wet season
treatment (DD), and an extended dry and wetter wet season treatment (ED) were
applied.    Tree water use and associated ecophysiological characters, such as the daily
whole tree transpiration ($E_L$), intrinsic water use efficiency ($WUE_i$), Huber values
($A_s$:$A_l$), and utilization proportions from different water sources were determined
during the period from October 2012 to September 2013.    For both tree species, no
significant difference in transpiration among the three treatments was observed in the
wet season, but a relatively stronger decrease of transpiration occurred under DD and
ED treatments during the later dry season.    Moreover, the higher transpiration of *M.*
*macclurei* and its advantage of utilizing the shallow water derived from light rainfall
under dry condition suggested that *M. macclurei* was more competitive in this
subtropical forest.    *M. macclurei* was inclined to adopt a drought-avoidance strategy,
whereas *S. superba* that could uptake deeper soil water was more likely to be drought
tolerant.    The different spatial and temporal patterns of water use, together with the
contrasting water use strategies, could partly explain the current coexistence of the
two tree species.    Their varying performance under different water conditions
implies possible shifts in species dominance within the forest community that were
potentially stimulated under future precipitation change scenarios from a long-term
perspective.



**Keywords:** sap flow, tree transpiration, plant coexistence, precipitation pattern
change, subtropical forest.


## 1 Introduction

Under the global climate changes, most ecosystems are predicted to be confronted with much severer environmental conditions, such as increasing aridity and frequency of extreme rainfall events, in the future (IPCC, 2013). Forest ecosystems are generally vulnerable to the increased intensity and frequency of drought events, which may reduce trees' survival, productivity and vitality (Allen et al., 2010; Cook et al., 2015). In this context, the variations in water availability and distribution may have profound influences on plant growth and survival at the ecosystem level (Drake and Franks, 2003; Nolan et al., 2018). To maintain high wood productivity and to counteract the effects of a changing climate on water availability for forest trees, it verifies the necessity of new and appropriate forestry management strategies in the future.

Mixed forests have been regarded as an alternative management practice to help forest ecosystems adapt to future climatic changes (Kelty, 2006; Grossiord et al., 2014), and the coexisting plants' capacity to exploit the spatiotemporally differentiated resources determines the degree to which resources are available for productivity in the ecosystem. In fact, while coexisting species compete with each other for resources, the complementarity effect in mixed forests implies that the coexisting species could seek unique ecological niches and use resources at different spatial locations or temporal segregations (Loreau and Hector, 2001). Attributed to the different tree traits, such as xylem trait, water use efficiency, root systems and stomatal regulations, the tree species coexistence is achieved at relatively small spatial



scales, especially under water-limited conditions (Forner et al., 2014; del Castillo et
al., 2016).    Sterck et al. (2011) has proposed that, in a water-limited ecosystem,
coexisting species may exhibit spatial or temporal resource partitioning and use water
more efficiently in order to maintain the forest growth and diversity.    Terrazas et al.
(2009) also verified that under a Mediterranean climate plant species with deeper
roots can make full use of groundwater resources, while those with shallow roots
mainly utilize episodic rainwater.    Other works also proposed some facilitation
processes, for example, the hydraulic lift by deep-rooted species favors neighbor
shallow-rooted species under water limited conditions (Prieto et al., 2012;
Rodríguez-Robles et al., 2015).    The result of Metz et al. (2016) strongly suggested
that the sensitivity of European beech to environmental factors depends on
neighborhood species, indicating that the development of mixed stands tends to be a
reasonable silvicultural strategy to mitigate drought effects on growth of *Fagus*
*sylvatica* stand.    In addition, the contrasting water use strategies of coexisting species
would also contribute to different responses of tree species to the moisture
environment, and consequently be beneficial for their coexistence.    Anisohydric
species displaying little stomatal regulation might suffer large fluctuations in
minimum leaf water potential, which are relatively drought-tolerant.    Isohydric
species, however, are often regarded as drought avoiders as they tend to occur in
mesic areas where they can avoid drought-induced hydraulic failure by way of strict
stomatal control and then relatively constant minimum leaf water potential
(McDowell et al., 2008).    For instance, as a water-saving and drought-avoiding



species, Mediterranean pines could share space and resources with drought-tolerant
and less conservative species such as evergreen oaks (Zavala et al., 2011).   However,
several recent studies have also shown that water deficit will increase the competition
for water resource in mixed forests (Grossiord et al., 2014), and consequently reduce
the potential benefits of species coexistence (Jucker et al., 2014).
Though coexisting plants often possess contrasting or complementary
resource-use strategies, related researches have been largely focused on arid and
semi-arid regions (Nolan et al., 2018; Forner et al., 2014), while studies addressing
the differential water use strategies of coexisting tree species in response to changed
seasonality of precipitation in subtropical moist areas are scant.   Despite the
abundant rainfall, many tropical areas with rich species have already experienced little
or no rain falls during dry seasons and upper soil layers might undergo severe drying
(Goldstein et al., 2008; Liu et al., 2010; Gao et al., 2015).   The unevenly distributed
precipitation might cause spatiotemporal changes in soil water availability, and thus,
would reshape the plant acclimation traits and water use (Gao et al., 2015; Ramírez et
al., 2009).   For example, del Castillo et al. (2016) reported that Aleppo pine and
holm oak shared the same hydrological niche when soil layers are well hydrated but
shifted to distinct water sources during periods of summer drought.   In addition,
adjustments of above- and below-ground biomass allocation in favor of the latter
would confer greater water transport capacity on a leaf area basis and help adapt to
the drier condition.   For example, Martin-StPaul et al. (2013) observed that the
transpiration of cork oak changed with the manipulated rainfall.   Their study also





showed no significant differences in the gas exchange parameters and leaf water
potential, while significant increase in ratio of branch sapwood area to leaf area ($A_\mathrm{s}$:$A_\mathrm{l}$)
was found for drought treatment plots, implying that higher $A_\mathrm{s}$:$A_\mathrm{l}$ could partially
compensate for the negative effect of decreased soil water availability, and thus ensure
a stable hydraulic continuum.    Moreover, as mentioned above, deep-rooted plants
have the advantage of exploiting relatively dependable water source that enables them
to survive long drought periods and to overcome seasonal water limitations (Dai et al.,
2015; del Castillo et al., 2016).    However, the soil water recharge from precipitation
pulses was the main water resource for shallow-rooted plants (Zapater et al., 2011;
Yang et al., 2011).    The different ability of plants to utilize available water of
different soil layers is apparently related to the precipitation pattern and intensity
during the growing seasons. In addition, species-specific seasonal patterns in
transpiration rates, together with the distinct ability to access water at different soil
depths, may lead to an alternation among competition, facilitation and niche
segregation patterns (Prieto et al., 2012; Rodríguez-Robles et al., 2015).    Therefore,
investigating the response of water use by coexisting plants to the soil moisture
dynamic changes are important to gain a deep understanding of the relationships
between precipitation patterns and plants growth.

In order to identify the physiological and ecological strategies of tree species

coexistence under the changing precipitation patterns, a field manipulation experiment
was conducted in a mesic forest located in Heshan County, Guangdong province,
South China.    Climate records of the Heshan County indicate increasing duration of



the dry seasons with more severe aridity intensity in recent years (Hu et al., 2018).
Main objectives of this study are 1) to investigate the changed traits and
spatial-temporal water use patterns of two dominant coexisting tree species (*Schima*
*superba* and *Michelia macclurei*) under the manipulated precipitation conditions in
this subtropical forest; 2) to determine the associated mechanism underlying the
different water use behaviors by examining tree transpirations and their responses to
changing environmental factors, the morphological adjustment of aboveground
biomass, the intrinsic water use efficiency, and the stable isotope composition of
xylem water and soil water.

**2 Materials and Methods**
**2.1 Site description**
Our study site is located in Heshan National Field Research Station of Forest
Ecosystem in the Heshan County, Guangdong Province, China (22° 41 ‘N, 112°
54′ E). Human disturbance had resulted in vegetation degradation in this region,
and an ecological restoration campaign attempting to reforest the degraded lands was
launched in the early 1980s. These man-made plantations have developed into
stable secondary broad-leaved evergreen forests after more than two-decade natural
growth (Hu et al., 2018). This area is dominated by a typical subtropical monsoon
climate, with long-term annual average air temperature of 19.1° C. The hottest and
coldest months are July (28.7° C) and January (13.7° C), respectively. Long-term
monitoring data show that the precipitation in this area has an annual average of



1813.6 mm and is unevenly distributed, with more than 70% of the rainfall occurring
from April to September (wet season) during which it is hot and humid.   It receives
less rainfall and is relatively cold and dry from October to March (dry season) (data
from http://www.cma.gov.cn/2011qxfw/2011qsjgx/).
The experimental site lies on a southeast-facing slope with an inclination of 16°.
Most of the plants are native species and approximately 25 years old, with *Michelia*
*macclurei* and *Schima superba* being the dominant tree species.   Forest density in
this experimental site is approximately 1019 trees per hectare.   The forest contains
an acrisol with a topsoil (0-20 cm) pH of 4.26, total nitrogen content of 1.2 g kg$^{-1}$,
organic carbon matter content of 24.2 g kg$^{-1}$, and available phosphorous content of 2.4
mg kg$^{-1}$ (Hu et al., 2018, Gao et al., 2015).
**2.2 Experimental design**
The Precipitation Seasonal Distribution Changes (PSDC) platform was established to
carry out the whole experiment.   We adopted four random blocks, 3 plots for each
block containing 3 treatments separately: "Blank control (BC)", "Drier dry and wetter
wet season (DD)¨, and "Extended dry and wetter wet season (ED)".   The DD
treatment was achieved by excluding 67% of throughfall during dry season (October
to March of the following year) using the under-canopy rain shelter devices to
simulate the drier condition under the dry season, whereas for the ED treatment, 67%
of throughfall was excluded in the spring (April-May) to simulate spring drought and
prolonged dry season.   To guarantee the equal total annual rainfall, approximately
equivalent amounts of excluded water were pumped into these plots several times



during wet seasons (from April to September for DD, and from June to September for
ED, respectively).   The irrigated water was pumped from a lake approximately 800
m away from the experimental site.   To minimize the interactions between the plots,
60-80 cm deep trenches were dug around the selected plots, and sufficient PVC plates
were buried to cut off the lateral surface runoff and interflow.   This operation could
also block the crosslinking among the sample tree roots.   More detailed information
about the facilities and the operations was comprehensively described in Hu et al.

(2018).

**2.3 Sap flow**
Two dominant coexisting tree species, *S. superba* and *M. macclurei* were chosen as
sample species for this research.   Sap flow of in total 24 *S. superba* and 39 *M.*
*macclurei* trees in all experimental plots was monitored from October 1, 2012 to
September 30, 2013.   The thermal dissipation probes (TDPs), with a length of 2.0 cm
and a diameter of 2.0 mm (Granier, 1987), were applied to measure trees' sap flux
density ($J_s$).   The upper probe was continuously heated by constant DC producing
power of 0.2 W, while the lower one was unheated and served as temperature
reference.   The self-made TDP probes were directly inserted into the xylem at a
height of 1.3 m above the ground on the northern side of tree trunk.   Sap flow
readings were recorded by the Delta-T data loggers (DL2e, Delta-T Devices, Ltd.,
Cambridge, UK).   The temperature difference between two probes was used to
calculate $J_s$ (g $H_2O$ $m^{-2}$ $s^{-1}$) according to the following equation (Granier 1987):
$$J_S = 119 \left( \frac{\Delta T_m - \Delta T}{\Delta T} \right)^{1.231} \tag{1}$$



where $\Delta T_m$ is the maximum temperature difference under zero-flux conditions, and $\Delta T$
is the instantaneous temperature difference.    To avoid the problem of radial variation
in $J_s$ with sapwood depth when integrating the measured sap flux values to whole tree
transpiration, the consistent relationship proposed by Pataki et al. (2011) for
angiosperm trees was applied as below:

$$J_i / J_o = 1.033 \times exp\left[-0.5\left(\frac{x-0.09963}{0.4263}\right)^2\right] \tag{2}$$

where $J_i/J_o$ is the ratio of sap flux at the actual to the outermost (2 cm in our study)
sapwood depth, and $x$ is the relative sapwood depth.    We first standardlized the sap
flux density and sapwood depth on the basis of $J_s$ and stem radius, then integrated the
obtained standardized function to get the standardized mean sap flux density ( $\overline{J}_{stan}$ ),
and consequently obtained the actual mean sap flux density ($J_s = \overline{J}_{stan} \times J_s/1$).    The
whole-tree sap flux was calculated by simply multiplying the mean sap flux density
with sapwood area ($E = \overline{J}_s \times A_s$, g s$^{-1}$).    To remove the effect of tree size on
transpiration, we adopted a normalized tree transpiration ($E_L$) expressed as $E/DBH$
following the proposal of Besson et al. (2014).

**2.4 Micrometeorology**

Photosynthetically active radiation ($PAR$), relative humidity ($RH$), air temperature ($T$),
and precipitation ($P$) were recorded hourly by a standard weather station 50 m away
from the experimental site.    Vapor pressure deficit ($VPD$) was calculated from $T$ and
$RH$ using the formula proposed by Campbell and Norman (1998) as follows:

$$VPD = a \times \exp(b \times T/(T+c)) \times (1-RH) \tag{3}$$

where $T$ is the air temperature (℃), $RH$ is the relative humidity of the air (%), and $a$, $b$





and *c* are constants with values of 0.611, 17.502 and 240.97, respectively.
Additionally, soil samples were periodically collected in the experimental plots to
measure the soil water contents (*SWC*) by gravimetric method.
**2.5 Tree biometric parameters**
Biometric parameters of the sample trees for sap flow monitoring were measured at
the beginning of the experiment.   Tree diameter at breast height (*DBH*) and tree
height (*H*) were measured using a *DBH* ruler and a Tandem-360R/PC altimeter
(Suunto, Finland), respectively.   We chose 20 trees of each species from the
surrounding area with different diameters to determine the sapwood area (*SA*), and
empirical equations between *SA* and *DBH* were established and then were used to
calculate the *SA* values for all sampled trees.   For wood density determination, we
used an increment borer to core the sapwood from six to seven trees outside the
experimental site.   The obtained wood cores were well wrapped by the wet towels
and placed in sealed plastic bags, then immediately transported to laboratory where
they were weighed by an electronic balance (Shinko, Japan, with an accuracy of
0.0001 g), and then dried to a constant weight at 80 ℃ in an oven to obtain the dry
weight.   The wood density values were calculated from the dry mass divided by
fresh volume.   The biometric parameters of the studied trees, including the diameter
breast height (*DBH*, cm), tree height (*H*, m), and sapwood area ($A_s$, $10^{-4}$ m$^2$) were
summarized in Table 1.
**2.6 Whole tree and branch $A_s$:$A_l$**
In this study, three to ten branches (20 cm) with 50-200 healthy leaves from each of





five replicate trees per species for each treatment were randomly sampled and
collected at the end of experiment.    All the leaf and wood samples of twigs were
oven dried at 80℃ to obtain a constant weight.    Branch barks were removed
carefully to measure the branch diameter and consequently to calculate the branch
sapwood area.    All leaves on each branch were scanned (Li-3000A, Li-Cor, Inc.,
Lincoln, NE) to calculate the branch $A_s$:$A_l$ (the ratio of sapwood area to leaf area).
The whole tree $A_s$:$A_l$ was obtained by the following procedures.    Firstly, we
calculated the values of leaf mass per area (*LMA*) according to the measured leaf
weight and the scanned leaf area mentioned above.    Then, we adopted the following
models to calculate the leaf biomass ($B_l$) (Gao et al., 2015):
*M. macclurei* : Log ($B_l$) = 0.5967 log ($DBH^2 \times H$) −1.0986 ($n$ = 4, $r^2$ = 0.96)        (4)
*S. superba* : Log ($B_l$) =0.7364 log ($DBH^2 \times H$) −1.7732 ($n$ = 4, $r^2$ = 0.99)            (5)
By combining the calculated data of *LMA* and $B_l$, we achieved the whole tree leaf area
($A_l$) and finally obtained the whole tree $A_s$:$A_l$.
**2.7 Water use efficiency**
The leaf-level intrinsic water use efficiency ($WUE_i$) was estimated by measuring
photosynthetic carbon isotope discrimination ($\Delta$) in bulk leaf tissue at the end of the
experiment (Farquhar et al., 1982).    As proposed by Farquhar et al. (1982), $\Delta$ is
inversely related to $WUE_i$ in C$_3$ plants, with $\Delta$ in bulk leaf tissue representing $WUE_i$
integrated over the time when carbon was assimilated.    The above-obtained dried
leaves described in the previous section were crushed and sieved through a 150 mesh,
and then used to measure the C isotopic signatures ($\delta^{13}$C, ‰) using Pee Dee



Belemnite (PDB) limestone and $N_2$ as the standards. Photosynthetic $^{13}C$
discrimination ($\Delta$) was then calculated as:
$$\Delta = \frac{\delta 13C_{atm} - \delta 13C_{plant}}{1 + \delta 13C_{plant}/1000}$$    (6)
where $\delta^{13}C_{atm}$ is the carbon isotope ratio of the atmosphere and assumed to be −8.72‰
(Gao et al., 2015). $WUE_i$ was calculated as:
$$WUE_i = \frac{C_a}{1.6} \times \left(\frac{27.5 - \Delta}{27.5 - 4.4}\right)$$    (7)
where $C_a$ is atmospheric carbon concentration (400 ppm), 27.5 (‰) is the
fractionation associated with enzyme reactions during $CO_2$-carboxylation, and 4.4 (‰)
is the fractionation during $CO_2$ diffusion through stomata.
*Stable isotope composition of xylem water and environmental water*
Different water samples for isotope analysis were collected from plant xylem water,
soil water, groundwater and rain at the end of the experiment (mid-September).
Suberized branch samples were collected from five selected trees for each treatment.
The green tissue and outer bark were carefully removed to prevent the isotopic
discrimination.    These pretreated branches were immediately cut into 1-cm long
segments, sealed in a glass vial, and stored at -20°C refrigerator after being
transported to laboratory.    Four rainfall samples were collected and analyzed for the
isotope analysis.    Soil samples at different depths (0-20, 20-40, and 40-60 cm) were
collected from each experimental plot.    Water from a small well near the
experimental plots was collected as the groundwater and kept in the laboratory at
0-5 °C.    The cryogenic vacuum extraction was used to extract water from soil and
branch samples, and the obtained water was filtered with microporous membranes



(pore size 0.45 μm) to remove solid organic matters (Ehleringer et al., 2000).    All the
prepared water samples were measured for the hydrogen/oxygen isotopic composition
using an isotope ratio mass spectrometer (Finnigan MAT253, USA).    Specifically,
the analyzer gave D and $^{18}O$ ratios relative to V-SMOW, and revisions were ± 1‰ and
± 0.2‰ for D and $^{18}O$, respectively.    D and $^{18}O$ compositions of water samples were
input in the IsoSource software V1.3.1 to quantitatively differentiate water in
branches absorbed from different water sources (Phillips and Gregg, 2003).    In the
process of calculation, mixtures were set to the hydrogen and oxygen isotopic
compositions of the branch water.    The increment and tolerance were set to 2% and
0.05%, respectively (Sun et al., 2018).
**2.8 Statistical analysis**
Differences of monthly $SWC$, whole-tree and branch $A_s$:$A_l$, and $WUE_i$ among tree
species and changed precipitation patterns were tested by the post hoc LSD test in the
SPSS software package (SPSS Inc. 2003).    Differences between the treatments were
considered statistically significant at $p < 0.05$.    To establish and compare the
correlations between whole-tree transpiration and $PAR$ or $VPD$, the linear model (y =
= $ax + b$) and exponential saturation model [y = $a \times (1-e^{-bx})$] were operated in Origin
8.0, where $a$ and $b$ are the fitting parameters.
**3 Results**
**3.1 Environmental factors**
As shown in Figure 1, the monitored environmental factors exhibited pronounced
seasonal variations.    The maximum monthly mean $T$ occurred in June with value of



27.72℃, while the minimum monthly mean $T$ was 13.83℃ and occurred in January.
Large variation was observed in daily $PAR$ values, ranging from 3.69 to 46.26 mol
$m^{-2}$ $d^{-1}$, and the monthly mean $PAR$ values during the whole experimental period
ranged from 16.44 (March) to 26.30 mol $m^{-2}$ $d^{-1}$ (June).   Total precipitation at the
research site during the experimental period was 2094 mm.   The precipitation was
unevenly distributed and occurred mainly between April and September, accounting
for approximately 84% of the annual total.   It was noticeable that the heaviest
precipitation with a value of 498.6 mm occurred in August, while the lightest
precipitation occurred in February with only 2.7 mm.   Difference in daily mean $VPD$
was remarkable between wet and dry seasons, reaching the peak (1.90 kPa) in
September and the lowest in March, respectively.   Monthly measured $SWC$ values
for the three manipulated precipitation treatments were shown in Figure 2.
According to the statistical analysis, the DD treatment possessed significantly lower
$SWC$ values for majority of the experimental months, with approximately 5%-30%
decline compared to BC and ED treatments, and no difference was observed between
the BC and ED treatments in the wet season.   Regarding the seasonal variations, the
highest $SWC$ values occurred in May for all three treatments, ranging from 26.0% to
31.0%.   Compared to the wet season, the average $SWC$ values decreased by
9.8%-13.7% in the dry season.
**3.2 Daily tree transpiration**
The daily normalized tree transpiration ($E_L$) of two tree species under dry (from
October to the next February), spring drought (from March to May), and wet (from



June to September) season was presented in Figure 3.    Generally, $E_L$ was higher in
wet season than in dry and spring drought seasons.    *M. macclurei* transpired more
water than *S. superba* under the same treatment for most sunny days, and it was more
significant during the periods of dry and spring drought seasons.    In terms of
temporal change, $E_L$ was relatively higher in wet and early dry seasons (October), but
showed a clear decline during later dry season, while increased and generally
maintained stable for the spring drought.    The changed precipitation pattern has
obviously posed an effect on tree transpiration.    Specifically, no significant
difference of transpiration for the three precipitation treatments was observed for both
tree species in the wet season, and such non-distinction in transpiration had continued
until later October.    During the dry season, trees in BC plots experienced a relatively
stronger transpiration (generally exceeded 40 kg day$^{-1}$ m$^{-1}$) than those under DD
treatments (mostly maintained at about 10-20 kg day$^{-1}$ m$^{-1}$ after November).
Differing from those in the wet and dry seasons, $E_L$ values of ED treatment were
significantly lower for both tree species than those of other two treatments during the
spring drought period.

To analyze the tree transpiration changes of two tree species with the changed

precipitation pattern, we averaged the daily tree transpiration and calculated the
decline percentages with the seasonal changes.    Compared to the wet season, the
transpiration of *M. macclurei* declined by 43% to 47% for the three treatments, while
the decline percentages for *S. superba* were from 33% to 46%, during the dry season,
and the DD treatment led to a largest decline in transpiration for both tree species



among the three treatments.    Similarly, the transpiration of *S. superba* and *M.*
*macclurei* under ED treatment during the spring drought period has decreased by
8.6% and 34%, respectively, with *M. macclurei* undergoing greater drop (26%-35%)
than *S. superba* (8%-28%) for the three different treatments.
**3.3 Water use efficiency and $A_s$:$A_l$ value**
As listed in Table 2, the water use efficiency ($WUE_i$) ranged from 64.8 to73.7 μmol
mol$^{-1}$ for *S. superba*, and 61.8 to 63.9 μmol mol$^{-1}$ for *M. macclurei*.    No distinct
precipitation treatment or species differences of $WUE_i$ were found, except a
significantly higher value for *S. superba* under DD treatment.    The branch and
whole-tree $A_s$:$A_l$, however, showed significant differences between two tree species.
To be specific, the branch and whole-tree $A_s$:$A_l$ of *M. macclurei* were 7.7% ~ 30.7%
lower than those of *S. superba* among the different rainfall treatments ($p < 0.05$).    It
is remarkable that the $A_s$:$A_l$ values of *M. macclurei* trees under the DD treatment
experienced the biggest drop (decreased by 30%), and the smallest decrease (with
values of 7.7% and 14%) under the ED treatment.    Whereas for the same tree species,
sampled trees in three different manipulated precipitation blocks shared similar
whole-tree $A_s$:$A_l$ values ($p > 0.05$).
**3.4 Proportions of water resources use**
Oxygen stable isotopes measurements and analyses by IsoSource model (Figure 4)
showed that trees obtained water predominantly from rainwater and soil water, which
generally account for more than 80% of xylem tree water use.    Normally, the
rainwater use of *M. macclurei* for BC and ED treatments was higher than that of *S.*



*superba*, but not for the treatment of DD.    The utilization of soil water by *M.*
*macclurei* trees showed no obvious treatment-difference.    However, attributed to the
full use of rainwater, the consumption of soil water by *S. superba* in DD plots (29%)
was relatively lower than that under the other two treatments (45.3% for BC, and
49.5% for ED, respectively).    In terms of soil water use depth, both tree species took
20.8% ~ 39.6% of water from a relatively deeper layers (40-60 cm soil layer and
groundwater), whereas the transpiration proportion obtained from shallow soil layers
water (0-40cm) for the different precipitation treatment plots accounted for 17.1%~
30.9%, and *S. superba* was inclined to use more deeper water and groundwater than
*M. macclurei*.
**3.5 Tree water use in response to *VPD* and *PAR***
Responses of $E_L$ to *VPD* and *PAR* for both species in dry, and spring drought and wet
seasons were presented in Figures 5-6, indicating that tree transpiration could be well
explained by *VPD* and *PAR*.    Significant linear relationships were established
between $E_L$ and *VPD* for the *S. superba* and *M. macclurei* ($R^2$ values ranged from 0.20
to 0.81, $p < 0.05$), except under BC treatment in wet season.    Normally, the slopes of
fitted lines in BC treatment were significantly higher than those in DD and ED
treatments, with values of BC > DD > ED in sequence.    During spring drought, a
much flatter change in daily transpiration with increasing *VPD* was observed in *M.*
*macclurei* of BC treatment.    For the DD and ED treatments, there was no significant
difference in the slopes of the fitted linear relationships for the three periods within
the same tree species.    We used the exponential saturation model to explore the





relationships between $E_L$ and *PAR* for all treatments.    As suggested by Gao et al.
(2015), parameter *b* might indicate the sensitivity of tree transpiration to the
environmental variables in the exponential saturation model.    Compared with BC
and DD treatments, tree transpiration under ED treatment for both species generally
showed less sensitivity to the increasing *PAR*, especially under dry season.    Further,
variations in parameter *b* could not be ignored, with values ranging from -6.89 to 0.08
for different treatments.    Though no obvious change pattern was observed for the
parameter *b* in the relationships between tree transpiration and *PAR*, the changes of $E_L$
with increased *VPD* still indicated that the sensitivity of *M. macclurei* was slightly
higher than that of *S. superba*.
**4 Discussion**
**4.1 Transpiration**
The results indicated that tree water utilization varied with time and tree species at the
experimental site.    Changed climatic indices are the main reasons for the temporal
variation of tree water use, as partly supported by the well-established relationships
between $E_L$ and *VPD* or *PAR* (Figure 5 and 6).    With more precipitation, higher *SWC*,
*VPD*, and *T* values, both tree species undoubtedly transpired more water during the
wet season.    Despite sufficient precipitation, tree transpiration still experienced a
decrease from March to May, even under the BC plots, which is mainly due to the
cloudy/rainy days and lower *VPD* or *PAR*.    It is noticeable that the transpiration in
October for both species remained at a relatively high level, which could be attributed
to the correspondingly higher evaporative demand and *PAR*.



Tree hydraulic characters and biometric parameters could explain the diverse tree
water use (Zinnert et al., 2013; Seyoum et al., 2014). For example, *S. superba*
possessed a relatively higher wood density and a less transpired water than *M.*
*macclurei* (Table 1 and Figure 3). Similar results were also reported by Köcher et al.
(2013), which demonstrated that tree species with lower wood density might have the
ability to utilize more water when transpiration demands are high than species with
higher wood density. Since the hydraulic conductivity is conversely related to
sapwood density (Pratt et al., 2007), the lower wood density of *M. macclurei* favored
a higher hydraulic conductivity, partly explaining why *M. macclurei* had the higher
transpiration quantity during most experimental time. Results indicated that the *S.*
*superba* had a significantly larger Huber value ($A_s$:$A_l$) (Table 2), which means this
species would be less access to water and can further reduce the risk of xylem
cavitation (Zolfaghar et al., 2014). Similar results, i.e., larger Huber values but less
transpired water, were also reported in Nolan et al. (2018), indicating that *S. superba*
was more likely to be drought-tolerant. As a stable and reliable indicator, xylem
water $\delta^{18}$O values can be regarded as an integrated estimate of water uptake by roots,
and it could help to distinguish the main water source used by a plant by comparing
them with those of potential water sources (Jackson et al., 1999; Liu et al., 2010).
We compared the xylem water $\delta^{18}$O values between *S. superba* and *M. macclurei*
(-5.80 ± 0.02‰ and -5.66 ± 0.28‰, respectively) and presented the water use
proportion in Figure 4. The results suggested *M. macclurei* used less groundwater,
but consumed more water from the shallow soil (0-60 cm soil depth) than *S. superba*.





Combined this water use proportion with the hydraulic characters (for example, Huber
value, stem wood density, etc.), the water relations of *M. macclurei* and *S. superba* are
consistent with drought avoidance and drought tolerance strategies, respectively.

**4.2 Influence of changed precipitation patterns on water use of coexisting trees**

As illustrated in Figure 3, the manipulated precipitation has significantly changed the
transpirations of both *M. macclurei* and *S. superba*.    Similar reduction of tree
transpiration following precipitation exclusion was also reported in other studies
(Besson et al., 2014; Pangle et al., 2015).    The significant decrease of soil water
content and the associated water availability were considered as the most direct reason
for the decrease of tree water use (Figure 2 and 3).    Furthermore, precipitation was
also a crucial limiting factor of $WUE_i$ (Battipaglia et al., 2014).    Scanlon and
Albertson (2004) pointed out that $WUE_i$ changed along the aridity gradient and
increased as precipitation decreased.    Moreno-Gutierrez et al. (2012) also stated that
many drought tolerant plants have increased $WUE_i$ compared to drought avoiding
plants.    In this study, an obvious increase of $WUE_i$ of *S. superba* in DD treatment
might indicate its better ability to cope with drought and ensure their own growth.
Moreover, under the conditions of water scarce, drought stress is the main influencing
factor on plant survival and growth.    Various mechanisms, including controlling
growth rate, adjusting leaf area index, increasing $WUE_i$, and uptaking water from deep
soil, would help plants adapt to this stress (Lévesque et al., 2014; Nock et al., 2011;
Sun et al., 2011).    In our study, the utilization of water from distinct soil layers by the
two tree species was observed under relatively drier condition.    The difference in



root biomass distribution of *M. macclurei* and *S. superba* may be the possible reason
for the different water use proportion.    According to Hu et al. (2018), *S. superba* and
*M. macclurei* allocate approximately 47% and 72% of the total root biomass to the
shallow soil layers, respectively.    This could also explain the higher transpiration rate
of *M. macclurei* than that of *S. superba* even during dry and spring drought periods, as
the less and lighter rain events that only kept the soil upper layer moist could render
*M. macclurei* convenience of obtaining shallow layer water, while *S. superba* had to
turn to deeper soil water by way of allocating more root biomass to the deeper soil
layers.

**4.3 Implications**

Availability of water can influence species composition and structure in many
ecosystems as well as species distribution of vegetation zones (Corbin et al., 2005;
Liu et al., 2010).    Our result that the *M. macclurei* maintained a higher transpiration
even under the relatively dry condition suggests its advantage under the present
environment, but it would face the risk of embolism in severe long-term drought due
to its relatively more root biomass allocation in shallow soil, lower wood density and
$A_s$:$A_l$ values.    In contrast, with more root biomass allocated in the deep layer, higher
Huber values, and higher wood density, *S. superba* might be drought tolerant and less
prone to xylem embolism (McDowell et al., 2008).    Additionally, the sensitivity of
tree transpiration to meteorological factors such as *VPD* and *PAR* could be indicated
by the slopes of the established fitting functions (Figure 5 and 6).    As proposed by
Sala et al. (2010), lower slopes implies a less increasing extent of water transpiration



following the increasing *VPD* or *PAR* under the DD and ED treatment, suggesting a
potential of smaller increasing extent of carbon uptake due to the stomatal closure
when potential drought stress happens.    Considering the importance of stomata
sensitivity for tree's growth, a higher transpiration rate under low *VPD* and higher
light demands are regarded as adaptive characteristics of the pioneering successional
tree species for ecological restoration (van Gelder et al., 2006), and our results also
proved that the *M. macclurei* was more sensitive to the environmental variations than
and therefore possessed a competitive advantage over *S. superba* under current
climatic condition in this moist forest.    These different water use strategies allow the
coexisting species to exploit resources differentially and can partially explain the
current coexistence of both species.    However, changes in the length and intensity of
drought events could lead to alternation in the dominance of tree species.    This
becomes particularly important for lower subtropical ecosystems in South China,
where it has experienced considerable chages of precipitation patterns in the recent
decades (Cao et al., 2012).    From this point, a chronic, prolonged drought could have
a stronger negative effect on *M. macclurei* than on *S. superba*, since hydraulic failure
would become a serious threat under long droughts.    Therefore, we might expect that
their current coexisting relations be altered under the potential future changes in
precipitation pattern.
**5 Conclusion**
Manipulated precipitation changes including drier condition and changed precipitation
seasonality have a species-specific impact on water use of dominant tree species in



the subtropical evergreen broad-leaved forest.   During the experimental period,
normalized daily transpiration was generally higher in wet season than those under
dry and spring drought condition.   *M. macclurei* that distributes more root biomass in
shallow soil layers transpired more water than *S. superba* even under dry/spring
drought period, implying that the shallow soil layer still does not experience the
drought stress under the current climate conditions, and thus the advantage of
acquiring shallow water for *M. macclurei* is guaranteed.   The manipulated
precipitation exclusion significantly reduced the transpiration for both tree species,
and a greater decrease of $E_L$ was observed for *M. macclurei* than for *S. superba* under
the drier conditions.   Though no significant difference in branch and whole $A_s:A_l$
values was induced by the precipitation exclusion, the measured oxygen stable
isotopes showed utilization of distinct water resources for the two studied tree species,
with *M. macclurei* preferring to a shallow soil water, and *S. superba*, however, being
more inclined to a deeper soil water.   Linear relationships between $E_L$ and *VPD*
established for both species under different treatments further explained the
species-specific water use under the changing water conditions.   Our findings have
emphasized the importance of current changing precipitation patterns in subtropical
moist zones for the coexistence of maturing individuals with different functional
types.

*Author contribution statement.* ZP, SWJ, GJG and OYL conceived and designed the
experiments. GJG, ZP, RXQ, ZLW, and NGY performed the experiments. OYL





analyzed the data and wrote the manuscript, ZP was involved in the revision of the
manuscript, other authors provided editorial advice.
*Competing interests.* - The authors declare no conflict of interest.
*Acknowledgments.* We would like to thank the staff in Heshan National Field
Research Station of Forest Ecosystem, and particularly thank Mr. Zhipeng Chen who
helped with instruments maintenance. This study was financially supported by
National Natural Science Foundation of China (41630752, 31700334, 41172313,
31425005) and National Key Research and Development Program
(2016YFC0500106-02).

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





**Table 1.** Biometric characters of the trees selected for sap flow measurement

| Tree species | S. superba | | | M. macclurei | | |
| --- | --- | --- | --- | --- | --- | --- |
| Experimental treatments | BC | DD | ED | BC | DD | ED |
| N | 8 | 10 | 6 | 15 | 12 | 12 |
| Height (m) | 7.0 ± 0.5 | 7.0 ± 0.5 | 6.5 ± 0.5 | 9.4 ± 0.6 | 13.6 ± 0.5 | 9.5 ± 0.5 |
| DBH (cm) | 12.9 ± 1.2 | 14.0 ± 0.9 | 13.7 ± 1.2 | 17.5 ± 1.4 | 19.0 ± 1.4 | 17.9 ± 1.3 |
| Sapwood area (cm²) | 116.4 ± 2.5 | 133.2 ± 6.8 | 133.3 ± 6.1 | 203.0 ± 1.1 | 234.3 ± 29.0 | 205.7 ± 8.2 |
| Projected crown area (m²) | 12.8 ± 2.6 | 13.1 ± 1.8 | 14.4 ± 2.1 | 30.1 ± 5.5 | 31.5 ± 4.1 | 25.7 ± 3.0 |
| Wood density (g cm⁻³) | | 0.61 ± 0.03 | | | 0.53 ± 0.03 | |

BC: an ambient control treatment, DD: a drier dry season and wetter wet season treatment, and ED an extended dry season and wetter wet season treatment.
*DBH*: tree diameter at breast height.





**Table 2.** The intrinsic water use efficiency ($WUE_i$, µmol $CO_2$ mol$^{-1}$ $H_2O$), branch and
whole-tree $A_s$:$A_l$ values (mm$^2$ cm$^{-2}$ x 10000) for *S. superba* and *M. macclurei*

| Treatment | BC | DD | ED |
|---|---|---|---|
| *S. superba* | | | |
| $WUE_i$ | 66.0 ± 3.1 a | 73.7 ± 3.5 b | 64.8± 4.0 a |
| Branch $A_s$:$A_l$ | 1.68 ± 0.16 cd | 1.86 ± 0.18 d | 1.53 ± 0.09 bc |
| Whole tree $A_s$:$A_l$ | 3.55 ± 0.50 bcd | 3.70 ± 0.41 cd | 3.80 ± 0.42 d |
| *M. macclurei* | | | |
| $WUE_i$ | 61.8 ± 2.6 a | 62.6 ± 5.0 a | 63.9 ± 3.8 a |
| Branch $A_s$:$A_l$ | 1.35 ± 0.05 ab | 1.20 ± 0.06 a | 1.42 ± 0.05 abc |
| Whole tree $A_s$:$A_l$ | 3.11 ± 0.65 ab | 2.83 ± 0.38 a | 3.23 ± 0.68 abc |

BC: an ambient control treatment, DD: a drier dry season and wetter wet season treatment,
and ED: an extended dry season and wetter wet season treatment.





**Figure Captions:**

**Figure 1.** Daily mean values of (a) photosynthetically active radiation (*PAR*), (b) temperature

(*T*), (c) vapor pressure deficit (*VPD*), and (d) precipitation (*P*) during the experimental period

(from October 1, 2012 to September 30, 2013).

**Figure 2.** Monthly soil water content under treatment of BC: an ambient control treatment,

DD: a drier dry and wetter wet season treatment, and ED: an extended dry and wetter wet

season treatment.

**Figure 3.** Daily water transpiration of *M. macclurei* (a, c and e, respectively) and *S. superba*

(b, d and f, respectively) during the dry season (the upper, from October, 2012 to February,

2013), spring drought (the middle, from April to May, 2013), and wet season (the bottom,

from June to September, 2013). Missing data were due to instrument failure or power-off. BC:

an ambient control treatment (open circles), DD: a drier dry and wetter wet season treatment

(open triangles), and ED an extended dry and wetter wet season treatment (half-filled

squares).

**Figure 4.** Proportions of the different water sources used by *S. superba* (left) and *M.

macclurei* (right) under different treatments. BC: an ambient control treatment, DD: a drier

dry and wetter wet season treatment, and ED an extended dry and wetter wet season

treatment.

**Figure 5.** Response of average daily water transpiration to average daily vapor pressure

deficit (*VPD*) for *M. macclurei* (a, c and e, respectively) and *S. superba* (b, d and f,

respectively) during the dry season (the upper), spring drought (the middle), and wet season

(the bottom). BC: an ambient control treatment (open circles and black lines), DD: a drier dry





and wetter wet season treatment (open triangles and green lines), and ED an extended dry and
wetter wet season treatment (half-filled squares and blue lines). All displayed fitted lines
showed significant linear regressions ($p < 0.05$).
**Figure 6.** Response of average daily water transpiration to daily *PAR* for *M. macclurei* (a, c,
and e, respectively) and *S. superba* (b, d and f, respectively) during the dry season (the upper),
spring drought (the middle), and wet season (the bottom). BC: an ambient control treatment
(open circles and black lines), DD: a drier dry and wetter wet season treatment (open triangles
and green lines), and ED an extended dry and wetter wet season treatment (half-filled squares
and blue lines). All displayed fitted lines showed significant linear regressions ($p < 0.05$).





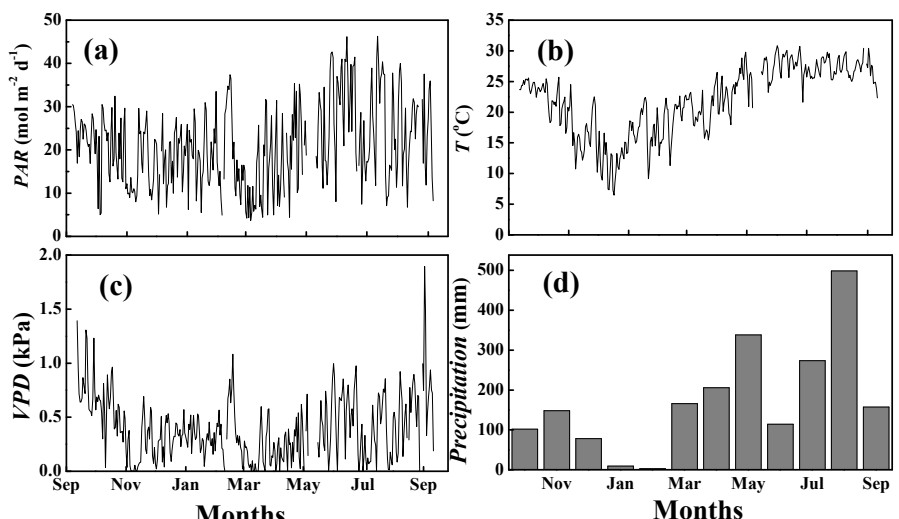


**Figure 1**








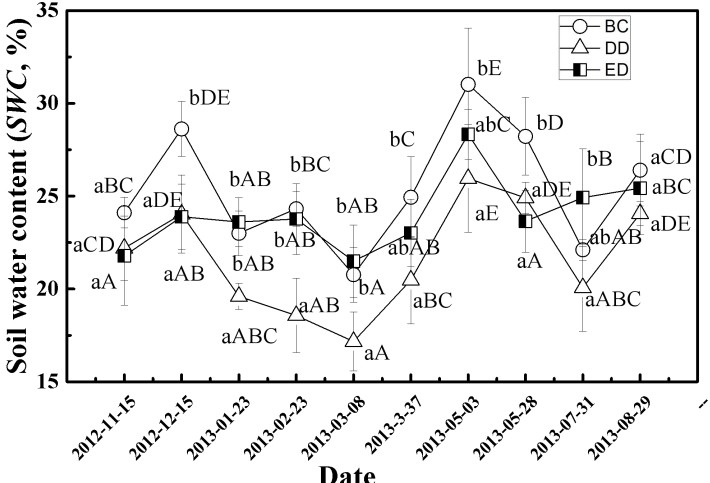


**Figure 2**






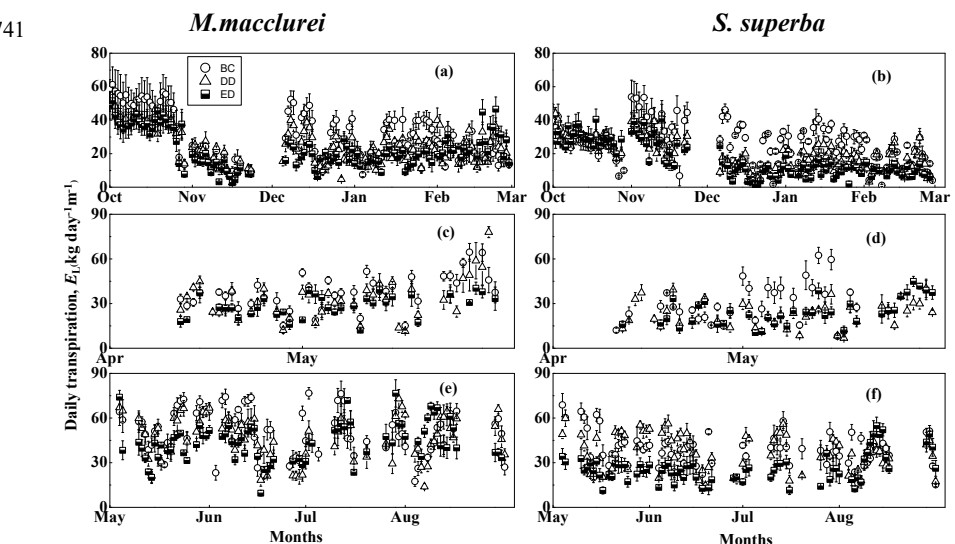


**Figure 3**






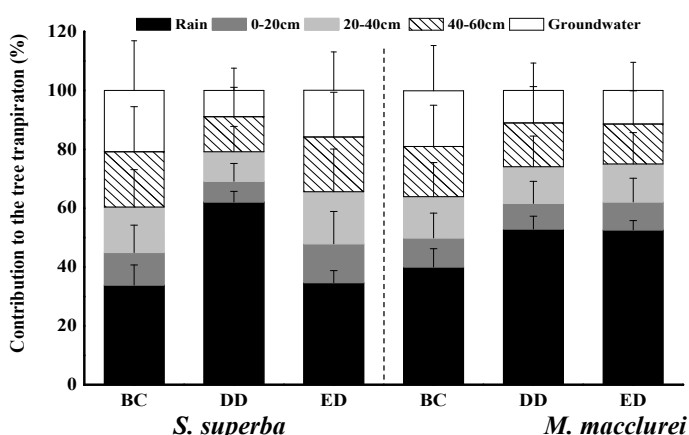



**Figure 4**





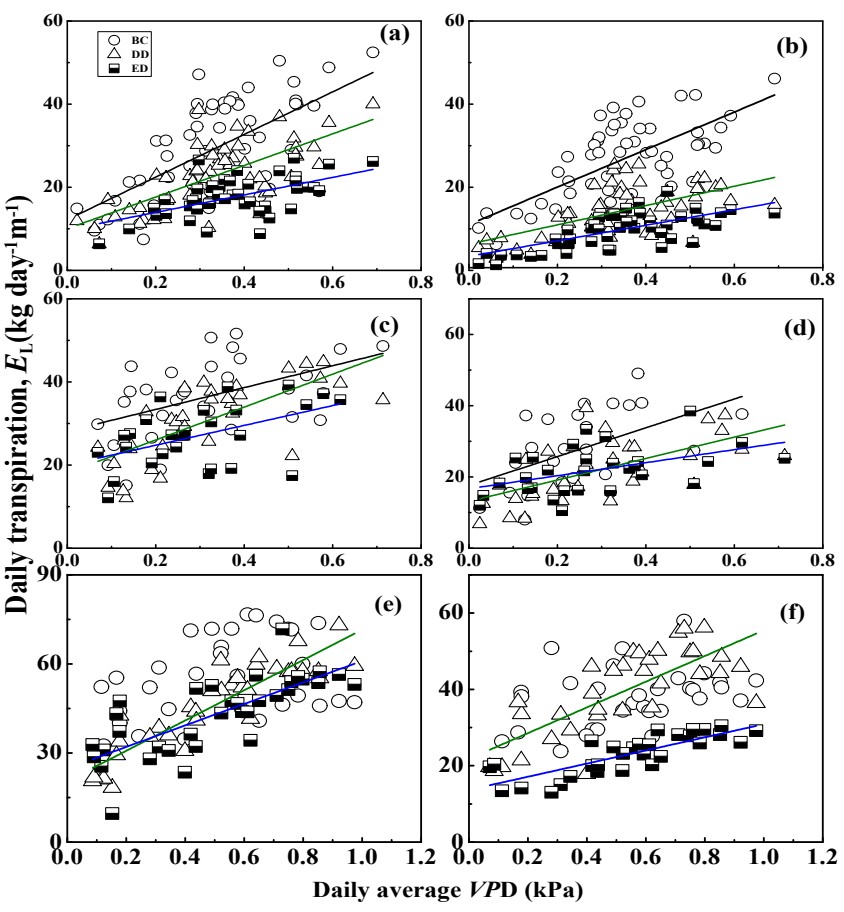


**Figure 5**



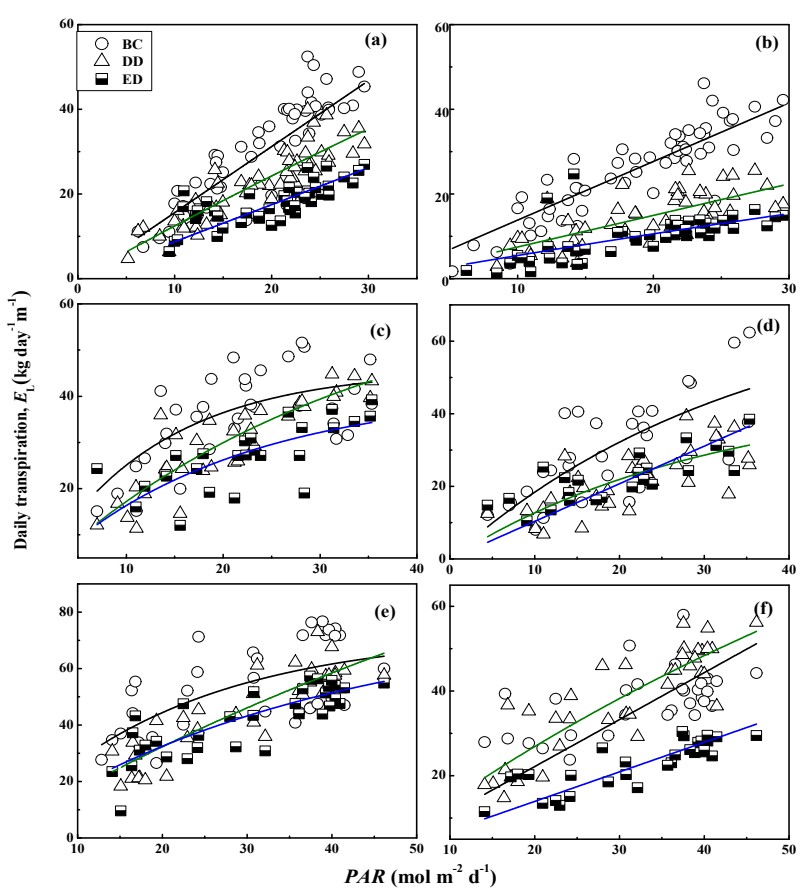


**Figure 6**