# Peer review of "Reshaped acclimation traits of dominant tree species under manipulated rainfall would alter their coexisting relation in a low-subtropical secondary evergreen forest"

_Biogeosciences, 2019_

## Referee Comment (RC1) · Anonymous Referee #1 · 24 Oct 2019

General comments: In this manuscript, Ouyang et al. present the results of a precipitation manipulation experiment in which the dry season was exacerbated or lengthened (along with a compensating increase in wet season water supply). The authors report a number of traits for the two dominant tree species at their experimental site, along with species-specific transpiration and water-use patterns. They conclude that the two dominant species show contrasting water-use strategies and their findings have important implications for the survival of these species under a changing climate regime. The experimental setup and amount of data collected is impressive, but the manuscript has

some key issues that I believe preclude publication at this time. Below, I have listed some broader feedback about the manuscript, followed by smaller line by line comments.

1) In general, I think the most novel aspect of this manuscript is the water uptake depth data. However, I think there is significant methodological detail missing about how water uptake depth was partitioned and there seems to be no statistical analysis comparing these data across treatments or species. I also think it should be mentioned that water samples were collected during a very limited time frame and that plant water use could have shifted during the experiment, as could precipitation 18O. Further, the authors make claims about these data (primarily differences between species), that look unfounded to me but could potentially be shown statistically. In sum, I don't think the water uptake depth analyses are valid as is. Without the water uptake depth data, I think the manuscript is fairly simple, merely showing the effects of the precipitation manipulation on transpiration, along with a few traits. 2) Related to the above point, I think there are some large gaps in the methods that make a solid interpretation of the results difficult. In particular, I think the manuscript is missing detail on leaf 13C sampling, lacks a discussion on the age of the bulk material they sampled for 13C analysis and what this means for their interpretations, neglects description of their water uptake depth model, and lacks some additional smaller details on sampling that I've highlighted below. 3) In many places throughout the manuscript, the authors claim their results speak to the competitive ability of these species using words like "competition", "success", "coexistence", "tolerance", etc. I do not think the data support these claims since the manuscript merely shows patterns in water use between the two species and some additional traits. I think this language should be toned down and I've highlighted areas where this issue came up. 4) Certainly, this is not a big issue at this stage, but I think the manuscript could use some significant editing for grammar and sentence structure.

Line by line comments: L1-2: I would really recommend altering the title. I do not

really think the study pertained to "acclimation traits", nor were any traits "reshaped". Further, I think that saying they "altered their coexisting relation" is not supported by the data (my point #3 above). L27: See #3 above about claims of competitiveness. L29-30: How are you defining drought tolerance? It seems like the two species had fairly similar responses to the precipitation manipulation in terms of transpiration and there was no evidence that either species was water stressed (point 3# above). L31: See #3 above about the coexistence terminology. L78-80: I do not think it is well supported in the literature that isohydric species tend to occur in mesic areas. They can and do exist most everywhere. L128-135: It seems like #1 and #2 are really similar in the sense that they both describe traits and water-use patterns. Maybe the authors could separate these objectives out into one about tree water use (and water uptake depth), and one about how traits mediate tree water use patterns. L161-164: I think it would be very helpful for the reader if there was a simple diagram (or perhaps labels added on to a time series figure) showing when precipitation was excluded or added back in for each treatment. L178: In order to interpret the effects of the precipitation manipulation, there needs to be information about how the experimental year's climate related to average site climate. For example, if this was a really wet year, there may be no reason to expect significant treatment effects in the first place. L216: What depth were these samples collected at? L252-290: I think some text should be added (either in the methods or discussion) that clarifies what the 13C signal would represent. If these leaves had been around prior to the experiment, their bulk tissue 13C would incorporate the 13C signal from climatic conditions at the time of leaf expansion, the carbon used to make those leaves, and any dynamics influencing non-structural carbohydrates since leaf expansion. Any information on the life span of these leaves would help here, or simply a caveat that the 13C signal could be complicated. L275: When were rainfall samples collected? Were these multiple samplings or were there 4 replicates of one rainfall event? If it is the latter, I don't think you can assume that this rainfall event is representative of all rainfall. L286: Please describe in detail what IsoSource is and the methodology behind how it partitions water uptake depth. L343-350: What are

these percentage reductions in comparison to (i.e., what counted as dry versus wet seasons)? L364-378: As mentioned in point #1, this sections needs some statistical analysis to be able to draw any conclusions from the data. L390-391: Is this saturation model warranted for the data considering that most of the responses seem to be fairly linear? Is there a first principles reason to expect a saturation relationship? L424: I'm confused as to how Huber value can be used to understand how much water a species has access to. L427: See point #3 above regarding the "drought-tolerant" terminology. L433-434: I'm not sure this is supported by the data (especially since there are no stats). The bars seem to be similar in size and the error bars seem to overlap. L489-490: I'm not sure this claim is supported since the treatment effects seemed to be fairly similar in the two species. Perhaps specify or cut this sentence? All figure and tables: Please specify what the +/- means in the tables (standard error, deviation?), what the lettering notation indicates, and what error bars represent. Fig. 1: During what hours were these daily values calculated? It might be more relevant to present mean daytime PAR and VPD. Fig. 3: I think the clarity of this figure could be improved. In general, it is hard to parse out trends due to the experiment since the points are so close together. It is also hard to interpret the panels for each species since they encompass different and overlapping time points. Perhaps clarify what manipulation was occurring in each panel, or maybe make one longer time series graph with all the treatment times labeled?

---

## Referee Comment (RC2) · Anonymous Referee #2 · 6 Dec 2019

Reshaped acclimation traits of dominant tree species under manipulated rainfall would alter their coexisting relation in a low-subtropical secondary evergreen forest

Authors: Ouyang et al.

Summary This study examined various factors, including transpiration rates, intrinsic WUE, sapwood to leaf area ratios, and water source use of two tropical trees under two precipitation exclusion treatments. The study found differences in transpiration rates between the wet and dry periods, and that the exclusion treatments reduced

transpiration rates in both species. The study also found a difference in soil water use between the two species, with one using more shallow and the other using deeper.

General Comments This manuscript tackles an interesting question related to how decreases in precipitation influences tropical forests, since more intense wet and dry periods are predicted for the future. The combination of measurements (transpiration, WUE, source water use) were all important in understanding what the mechanisms were for different water use strategies. However, there were several weaknesses with the paper that made it difficult to recommend for publication. 1) The objectives of this study were not particularly exciting. Why was the main objective simply to examine different water use strategies by different tree species? This study utilized two precipitation exclusion treatments, ED and DD, and these different deliveries of moisture provided a unique opportunity to ask more targeted questions and/or hypotheses. This was a missed opportunity. The second objective was to examine if the mechanism responsible for the differences were due to transpiration, morphological adjustment, or WUE. Again, this is not particularly exciting because the answer to this question is yes, all of these likely differ (or not differ) between the two species. The more important question is, if they do differ (or not differ), why? Can you provide some hypotheses on how you might think each species may respond both physiologically as well as morphologically to the different treatments? 2) This manuscript would also benefit a lot by providing a more clear picture of when measurements were made. For example, the statistical analysis section seemed to imply that analyses were made at monthly time intervals, but the method section described making many of the collections at the end of the experiment. It wasn't clear how monthly time scale analyses could be made if there were only one set of collections. 3) The Results section lacked reporting of the statistical analyses. This needs to be addressed. Without the statistical outputs, it's hard to evaluate if any of the findings were true. 4) Despite having two different precipitation exclusion treatments, the results are not discussed at all in the Discussion section. 5) The Discussion section should be improved to provide a cohesive story about whether or not species differences matter, or it different water treatments matter.

I found this discussion difficult to follow because it read as many disjointed sentences highlighting when the two species behaved differently and when they behaved similarly.

Specific Comments

Title: I recommend changing the title of this manuscript. The idea of "reshaped acclimation traits" is not very clear, and neither is "would alter their coexisting relation." The title should represent the key findings of the study and I don't think either of these phrases capture that.

Line 32. The idea that physiological differences alone would explain shifts in species composition is a bit of an overstretch. What about seedling recruitment and seedling success? I would recommend ending the abstract based on the findings.

Line 41. Change to "...with much MORE SEVERE environmental conditions..."

Line 48. Change to "...for forest trees, new and appropriate forestry management strategies ARE NEEDED in the future."

Line 98. The paragraph begins with highlighting the fact that studies linking changes in rainfall and vegetation water use are typically addressed in semi-arid and arid ecosystems and that tropical areas are largely ignored. However, the citation of del Castillo (2016) for Aleppo pine refers to Mediterranean climates. Another citation would be more appropriate here.

Line 112. Insert "...the soil water recharge from SHALLOW precipitation..."

Line 128. What specific traits do "changed traits" refer to here?

Line 134. "...and the stable isotope composition of xylem and soil water." Isn't the use of stable isotope part of examining spatial-temporal water use patterns (from objective #1)? Using isotopes to trace water is a tool, not an underlying mechanism.

Line 144. Insert "...after more than two-decadeS OF natural growth."

Line 149. "...and is evenly distributed, with more than 70% of rainfall occurring from April to September..." This sounds contradictory. How can rainfall be even distributed if more than 70% falls during the wet season? Is this referring to spatial distribution somehow?

Line 165. I'm not sure I understand the rationale in the precipitation exclusion treatments between ED and DD. If the dry period is from October to March and the ED period reduces precipitation from April to May, how is this different than DD?

Line 167 also says "...whereas for the ED treatment, 67% of throughfall was excluded in the spring (April-May) to simulate spring drought and prolonged dry season." This may extend the dry season by excluding precipitation into May, but was precipitation not altered during winter (October to March)? Some additional text explaining the rationale for the precipitation treatments would help clarify this.

Line 216. At what depths were soil samples collected for SWC measurements?

Line 239. Change to "...BRANCH BARK WAS removed..."

Line 247 and 248. What do the "n=4" refer to? Were there only four trees used in this calculation of leaf biomass? If there were five replicate trees per species for each treatment, it's unclear where n=4 comes from.

Line 292. How were monthly differences in whole-tree and branch As:Al calculated? I was under the impression that As:Al was calculated at the end of the experiment. Also, were leaf tissue collected for d13C also collected monthly? Also, the previous section says that xylem water, soil water, and precipitation were collected at the end of the experiment. How can monthly differences then be calculated?

Line 293. Again, it's unclear if measurements/collections were made monthly or at the end of the experiment, so it's not clear if the LSD post hoc test is the best. If monthly measurements were made on the same sets of trees, a repeated measures analyses makes more sense.

Line 326. "M. macclurei transpired more water than S. superba. . ." I'm not sure what data support these findings

Line 338. Why was transpiration of trees from the ED treatments lower during the winter if rain was not excluded during the winter?

Lines 343-350. I would consider revising this section because it's hard to keep track of the decreases in transpiration rates between different seasons.

Line 352-355. Please report the statistics here.

Line 355-356. Please report statistics here.

Line 357. "To be specific, the branch and whole-tree As:Al of M. macclurei were 7.7% ∼ 30.7% lower than those of S. superba among the different rainfall treatments (p<0.005)." This is unclear – is the p-value saying that all M. macclurei treatments (BC, EE, ED) were significantly lower than S. superba? If so, why were the control, BC, treatments different?

Line 362. "Whereas for the same tree species, sampled trees in three different manipulation precipitation blocks shared similar whole tree As:Al values." What does this mean?

Line 367. "Normally, the rainwater use of M. macclurei for BC and ED treatments was higher than that of S. superba, but not for the treatment of DD." Please show the statistics.

Line 377. ". . .S. superba was inclined to use more deeper water and groundwater than M. macclurei." Please show statistics to support this.

Section 3.4 It's still unclear to me if xylem water was collected only once during the experiment (at the end), or if samples had been collected during winter, spring, and summer.

Section 3.5 Statistics are missing almost entirely from here. If slopes of one treatment

is higher than another treatment, please include the statistics. If the slope were not different, please show the statistics as well.

Discussion section. I recommend beginning with a summary of the key findings from this study before launching into the details of each type of measurement.

Line 424. "...species would be less access to water and can further reduce the risk of xylem cavitation..." I'm not following this argument.

Section 4.1 I don't see any discussion of how the different treatments influenced water use.

Table 2. The letters used to discern differences between BC, DD, and ED treatments are quite confusing. For example, why are different letters used for Branch As:Al (b, c, d) compared to WUEi (a, b). This almost implies that WUEi and As:Al were compared, when they clearly were not.

Figure 2. The letters used to show differences in SWC are too complicated. Please remove and just report the statistical findings. Why are lower case and capital letters both used? The Figure legend does not explain any of this.

Figure 3. Instead of splitting the daily E into dry, spring, and summer, I would plot daily E along one time axis. The way this figure is currently set up, the time intervals are different between panels a, c, e, and b, d, f. A better way is to highlight the different periods of the precipitation in one panel, and have M. Macclurei on top, and S. superba on the bottom. Also, why is ED lower during the dry season (Oct – Mar) if precipitation was not excluded during this time?

Figure 4. No statistics here. Why?

[Figure]

---

## Author Comment (AC1) · 7 Jan 2020

Reviewer 2: Summary This study examined various factors, including transpiration rates, intrinsic WUE, sapwood to leaf area ratios, and water source use of two tropical trees under two precipitation exclusion treatments. The study found differences in transpiration rates between the wet and dry periods, and that the exclusion treatments reduced transpiration rates in both species. The study also found a difference in soil water use between the two species, with one using more shallow and the other using deeper. General Comments: This manuscript tackles an interesting question related to

how decreases in precipitation influences tropical forests, since more intense wet and dry periods are predicted for the future. The combination of measurements (transpiration, WUE, source water use) were all important in understanding what the mechanisms were for different water use strategies. However, there were several weaknesses with the paper that made it difficult to recommend for publication. 1) The objectives of this study were not particularly exciting. Why was the main objective simply to examine different water use strategies by different tree species? This study utilized two precipitation exclusion treatments, ED and DD, and these different deliveries of moisture provided a unique opportunity to ask more targeted questions and/or hypotheses. This was a missed opportunity. The second objective was to examine if the mechanism responsible for the differences were due to transpiration, morphological adjustment, or WUE. Again, this is not particularly exciting because the answer to this question is yes, all of these likely differ (or not differ) between the two species. The more important question is, if they do differ (or not differ), why? Can you provide some hypotheses on how you might think each species may respond both physiologically as well as morphologically to the different treatments?

R: According to the suggestion, we have rewritten the objectives and provided the hypotheses: "Based on previous root sampling analyses that 47% and 72% of the total root biomass distributed in the surface soil layer (0-20 cm) for Schima superba and Michelia macclurei (two dominant coexisting tree species in our experimental site), respectively (Hao and Peng, 2009; Li, 1984), and combined with our preliminary survey on tree biometric characters, we hypothesized that 1) M. macclurei trees would transpired more water than S. superba tress, even under the relatively drier condition; 2) In response to the intensified drought or extension of dry season, the trees will improve their water use efficiency by increasing ratios of sapwood area to leaf area (M. macclurei), or by being promoted to utilize deeper soil water (S. superba). Therefore, the main objectives of this study are 1) to investigate the effects of manipulated precipitation conditions on spatial-temporal water use patterns of S. superba and M. macclurei in this subtropical forest; 2) to understand the potential mechanism for the

varied responses of tree transpiration to the changed precipitation patterns by examining the variations in morphological adjustment, such as (As: Al), the intrinsic water use efficiency, and the contributions of water resources to the tree transpiration" (Line 126-141)

2)This manuscript would also benefit a lot by providing a more clear picture of when measurements were made. For example, the statistical analysis section seemed to imply that analyses were made at monthly time intervals, but the method section described making many of the collections at the end of the experiment. It wasn't clear how monthly time scale analyses could be made if there were only one set of collections.

R: The time for the measurements in our study was presented in the section of Materials&Methods. As described in the section, the sap flow monitoring had been carried out continuously through the whole year with the recording interval of 10 minutes, We measured the soil water content (SWC) values monthly, but for the WUEi, Huber values, and the isotope analysis, collections and measurements were only carried out at the end of the experiment. (Line 244, 259, 282).

3)The Results section lacked reporting of the statistical analyses. This needs to be addressed. Without the statistical outputs, it's hard to evaluate if any of the findings were true.

R: According to the suggestion, we recalculate the proportions of water resources use based on the measured isotope data and display the statistical analysis in Figure 4. The corresponding results of statistical analysis are also described in the manuscript: "According to the statistical results, the utilization of rainwater and soil water of M. macclurei trees showed no significant treatment-difference, while the DD and ED treatments significantly decreased its utilization of groundwater (Figure 4). However, the changed precipitation pattern posed a significant influence on the water use proportions of S. superba from different water resources ($p < 0.05$)....Furthermore, the two dominant tree species shared similar water use proportion under the control condition, and M. macclurei used more soil water (0-60cm) than the S. superba under DD treatment. Comparatively, M. macclurei utilized more rainwater, while S. superba was inclined to make use of more groundwater under ED treatment ($p < 0.05$)." (Line 379-392)

4)Despite having two different precipitation exclusion treatments, the results are not discussed at all in the Discussion section.

R: We revised the Discussion section and added some contents according to the suggestion: "In this study, only S. superba experienced significant increase of WUEi under DD treatment, indicating its better ability to cope with intensified drought stress. Differing from S. superba, M. macclurei did not significantly change WUEi under DD and ED treatments. This less adjustment of WUEi together with its higher transpiration under relatively drier seasons implied an disadvantage for M. macclurei when facing water stress. Moreover, other mechanisms, including controlling growth rate, adjusting leaf area index, and uptaking water from deep soil, would help plants adapt to water scarce (Lévesque et al., 2014; Nock et al., 2011; Sun et al., 2011). In our study, compared with the BC condition, the DD and ED treatments did not significantly change the Huber values ($p > 0.$ 05), but did pose an obvious influence on the utilization of water from distinct water sources, especially for S. superba. Though the relatively higher use of rainfall water has decreased the use of groundwater under DD treatment, the prolonged dry condition still promoted S. superba to utilize deeper soil water (40-60 cm). From this point, a chronic, drought could have a stronger negative effect on M. macclurei than on S. superba. "(Line 472-487)

5)The Discussion section should be improved to provide a cohesive story about whether or not species differences matter, or different water treatments matter. I found this discussion difficult to follow because it read as many disjointed sentences highlighting when the two species behaved differently and when they behaved similarly.

R: Our writing ideas and framework are explained as follows: In the section 4.1, we

focused on the comparison of tree transpiration and the possible mechanisms between the chosen tree species under control condition (treatment of BC), while in the section 4.2, the point is the effect of changed rainfall patterns on tree transpiration, and how the indicators, such as the WUEi, Huber values, or water use proportions that varied with the treatments (DD and ED), help explain the changes in tree transpiration for both species. To follow the track of the ideas, we revised the Discussion section to avoid the structure shortcomings. Please see the revised text (in blue color) in the manuscript.

Specific Comments Title: I recommend changing the title of this manuscript. The idea of "reshaped acclimation traits" is not very clear, and neither is "would alter their coexisting relation." The title should represent the key findings of the study and I don't think either of these phrases capture that.

R: Yes, we have changed the title as "Species-specific transpiration and water-use patterns of two dominant coexisting tree species under manipulated rainfall in a lowsubtropical secondary evergreen forest"

Line 32. The idea that physiological differences alone would explain shifts in species composition is a bit of an overstretch. What about seedling recruitment and seedling success? I would recommend ending the abstract based on the findings.

R: The overemphasized part was deleted now in the manuscript.

Line 41. Change to ". . .with much MORE SEVERE environmental conditions. . ."

R: Done. (Line 39)

Line 48. Change to ". . .for forest trees, new and appropriate forestry management strategies ARE NEEDED in the future."

R: Done. (Line 47)

Line 98. The paragraph begins with highlighting the fact that studies linking changes in rainfall and vegetation water use are typically addressed in semi-arid and arid ecosystems and that tropical areas are largely ignored. However, the citation of del Castillo (2016) for Aleppo pine refers to Mediterranean climates. Another citation would be more appropriate here.

R: As you concerned, the reference we cited here refers to a Mediterranean climate, therefore, another study that focused on tree water use of a tropical seasonal rainforest were presented here. This sentence was also changed to "For example, Liu et al. (2010) reported that Pometia tomentosa tree seemed to tap water mostly from depths greater than 60 cm and groundwater owing to its deep taproot, while the fog water was an important source for its seedling growth at the peak of the dry season." (Line 95-98)

Line 112. Insert ". . .the soil water recharge from SHALLOW precipitation. . ."

R: Done. (Line 110)

Line 128. What specific traits do "changed traits" refer to here?

R: We have changed it by rewriting the OBJECTIVES in the manuscript. (Line 135-141)

Line 134. ". . .and the stable isotope composition of xylem and soil water." Isn't the use of stable isotope part of examining spatial-temporal water use patterns (from objective #1)? Using isotopes to trace water is a tool, not an underlying mechanism.

R: We have revised this part as "and the contributions of water resources to the tree transpiration..." (Line 135-141)

Line 144. Insert ". . .after more than two-decadeS OF natural growth."

R: Done. (Line 149)

Line 149. ". . .and is evenly distributed, with more than 70% of rainfall occurring from April to September. . ." This sounds contradictory. How can rainfall be even distributed if more than 70% falls during the wet season? Is this referring to spatial distribution somehow?

R: This was a mistake. We revised as "...and is unevenly distributed, with more than 70% of the rainfall occurring from April to September.." (Line 153)

Line 165. I'm not sure I understand the rationale in the precipitation exclusion treatments between ED and DD. If the dry period is from October to March and the ED period reduces precipitation from April to May, how is this different than DD?

R: DD means the drier dry season, we excluded 67% of throughfall during dry season (October to March of the following year), while ED means an extension of dry season: namely 67% of throughfall were excluded from April and May which originally belonged to the wet season and became dry. Please see the detailed information in Table 1 in the new version of manuscript.

Line 167 also says ". . .whereas for the ED treatment, 67% of throughfall was excluded in the spring (April-May) to simulate spring drought and prolonged dry season." This may extend the dry season by excluding precipitation into May, but was precipitation not altered during winter (October to March)? Some additional text explaining the rationale for the precipitation treatments would help clarify this.

R: The precipitation was not altered during winter, as we explained in the manuscript: "67% of throughfall in the spring (April-May) were excluded and the equivalent amounts of excluded water were pumped into these plots several times during wet seasons (from June to September for DD and ED treatments)". (See Line 172-178)

Line 216. At what depths were soil samples collected for SWC measurements?

R: We added the contents in the text: "Additionally, soil samples (0-30 cm) were monthly collected in the experimental plots to measure the soil water contents (SWC) by gravimetric method." (Line 222)

Line 239. Change to ". . .BRANCH BARK WAS removed. . ."

R: Done. (Line 245)

Line 247 and 248. What do the "n=4" refer to? Were there only four trees used in this calculation of leaf biomass? If there were five replicate trees per species for each treatment, it's unclear where n=4 comes from.

R: Yes, there were five replicate trees per species for each treatment, but here it refers to four non-sample trees used for the model establishment of leaf biomass.

Line 292. How were monthly differences in whole-tree and branch As:Al calculated? I was under the impression that As:Al was calculated at the end of the experiment. Also, were leaf tissue collected for d13C also collected monthly? Also, the previous section says that xylem water, soil water, and precipitation were collected at the end of the experiment. How can monthly differences then be calculated?

R: We are sorry for the misleading expressions. We measured the soil water content (SWC) values monthly, but for the WUEi, Huber values, and the isotope analysis, the collections and measurements were carried out only at the end of the experiment. Here this sentence was changed to avoid the misunderstanding: "Differences of SWC, whole-tree and branch As:Al, and WUEi among tree species..." (Line 309)

Line 293. Again, it's unclear if measurements/collections were made monthly or at the end of the experiment, so it's not clear if the LSD post hoc test is the best. If monthly measurements were made on the same sets of trees, a repeated measures analyses makes more sense.

R: The measurements of WUEi, Huber values, and the isotope analysis were carried out only at the end of the experiment (Line 259 and 282), and we used the One-way ANOVA followed by a post hoc test to test the differences among the different treatments in this study.

Line 326. "M. macclurei transpired more water than S. superba. . ." I'm not sure what data support these findings

R: The daily transpiration data of M. macclurei and S. superba was presented in Figure

3. The comparison might not be clear enough due to the relatively large amount of data. This sentence was changed to make it less controversial: " M. macclurei transpired more water than S. superba during wet and early dry seasons." (Line 346-347)

Line 338. Why was transpiration of trees from the ED treatments lower during the winter if rain was not excluded during the winter?

R: Sorry for having not presented the sentence clearly. What we actually expressed is " Differing from those in the wet and dry seasons, EL values of ED treatment were significantly lower for both tree species than those of BC and DD treatments during the spring drought period", rather than "during the winter" (Line 356-358)

Lines 343-350. I would consider revising this section because it's hard to keep track of the decreases in transpiration rates between different seasons.

R: Yes, we think this section did not contribute more to our findings in this study and therefore have deleted it.

Line 352-355. Please report the statistics here.

R: The statistics results were added in the text and in Table 3. "Compared with S. superba trees, M. macclurei had significantly lower branch As:Al values of under BC and DD treatments ($p < 0.05$). It is remarkable that the branch and whole As:Al values of M. macclurei trees experienced the biggest drop under the DD treatment, and the whole As:Al values showed a significant difference between two species only under DD treatment ($p < 0.05$). The manipulated precipitation treatments posed significant effect in branch As:Al for both S. superba and M. macclurei ($p < 0.05$), whereas no significant effect on whole-tree As:Al values was observed ($p > 0.05$)." (Line 363-371)

Line 355-356. Please report statistics here.

R: The statistics results were added in the text and in Table 3. (Line 363-371)

Line 357. "To be specific, the branch and whole-tree As:Al of M. macclurei were

7.7% âĹij 30.7% lower than those of S. superba among the different rainfall treatments (p<0.005)." This is unclear – is the p-value saying that all M. macclurei treatments (BC, EE, ED) were significantly lower than S. superba? If so, why were the control, BC, treatments different?

R: To separate the effects of tree species and precipitation treatment, we have redone the statistical analysis the results were listed in Table 3. Corresponding modification of the results was described in the text. "Compared with the S. superba trees, M. macclurei had significantly lower branch As:Al values of under BC and DD treatments (p < 0.05). It is remarkable that the branch and whole As:Al values of M. macclurei trees experienced the biggest drop under the DD treatment, while the whole As:Al values presented a significant difference between two species only under DD treatment (p < 0.05). The manipulated precipitation treatments posed significant effect in branch As:Al for both S. superba and M. macclurei (p < 0.05), whereas no significant effect on whole-tree As:Al values was observed (p > 0.05)." (Line 363-371)

Line 362. "Whereas for the same tree species, sampled trees in three different manipulation precipitation blocks shared similar whole tree As:Al values." What does this mean?

R: The meaning of this sentence was the manipulated precipitation treatments posed no significant effect on whole-tree As:Al values for both S. superba and M. macclurei, and here we revised this sentence to avoid confusion: "The manipulated precipitation treatments posed significant effect in branch As:Al for both S. superba and M. macclurei (p < 0.05), whereas no significant effect on whole-tree As:Al values was observed (p > 0.05)."(Line 368-371)

Line 367. "Normally, the rainwater use of M. macclurei for BC and ED treatments was higher than that of S. superba, but not for the treatment of DD." Please show the statistics.

R: The results of statistical analysis were added in Figure 4 and described in the

manuscript now: "Comparatively, M. macclurei utilized more rainwater, while S. superba was inclined to make use of more groundwater under ED treatment (p < 0.05)." (Line 390-392)

Line 377. ". . .S. superba was inclined to use more deeper water and groundwater than M.macclurei." Please show statistics to support this.

R: The results of statistical analysis were added in Figure 4 and described in the manuscript now: "Comparatively, M. macclurei utilized more rainwater, while S. superba was inclined to make use of more groundwater under ED treatment (p < 0.05)." (Line 390-392)

Section 3.4 It's still unclear to me if xylem water was collected only once during the experiment (at the end), or if samples had been collected during winter, spring, and summer.

R: We are sorry for the misleading. We have clarified the sampling time in the text "To identify the utilization of water resources by trees, we measured the hydrogen and oxygen isotopes (D and 18O) of xylem water and different water sources (rain, soil water from different soil layers, and groundwater) at the end of the experiment." (Line 280-282)

Section 3.5 Statistics are missing almost entirely from here. If slopes of one treatment is higher than another treatment, please include the statistics. If the slope were not different, please show the statistics as well.

R:Actually, we have done the statistics for the analysis of tree water use in response to VPD, and we added the description of statistics in the text: "To examine the differences in regressions for both tree species under manipulated precipitation treatments, we performed homogeneity of regression slopes and an analysis of covariance in SPSS software package (SPSS Inc., 2003). Differences between the treatments were considered to be statistically significant at p < 0.05." (Line 314-318). The statistic results of

the regression was also described in the text: "Normally, the slopes of fitted lines in BC treatment were significantly higher than those in DD and ED treatments ($p < 0.05$), with a value sequence of BC > DD > ED. During spring drought, a much flatter change in daily transpiration with increasing VPD was observed in M. macclurei of BC treatment. For the DD and ED treatments, there was no significant difference in the slopes of the fitted linear relationships for the three periods within the same tree species ($p > 0.05$)." (Line 398-404)

Discussion section. I recommend beginning with a summary of the key findings from this study before launching into the details of each type of measurement.

R: A summary of the key findings in Discussion section was added in the text: "Differing from previous studies that mainly focused on water-restricted habitats, we explored the variations in responses of water use of the two dominant tree species to the manipulated precipitation treatments in a subtropical forest ecosystem. Results support the first hypothesis that the M. macclurei trees usually transpired more water than S. superba trees due to the growth advantages in biometric characters of the former. Also, the manipulated precipitation exclusion significantly restrained the transpiration for both tree species, and the adjustments of Huber values, water use efficiency and the water uptake depth would partly responsible for the decreased tree transpiration. Our study indicated that, to cope with the potential seasonal drought in the future, the coexisting M. macclurei and S. superba trees will adopt drought avoidance and drought tolerance strategies, respectively." (Line 416-426)

Line 424. ". . .species would be less access to water and can further reduce the risk of xylem cavitation. . ." I'm not following this argument.

R: To make it easy to understand, we revised this sentence as : "Results indicated that S. superba had a significantly larger Huber values (As:Al) (Table 2), possibly meaning a less investment in leaf biomass and a better efficient transport system (Zhu et al., 2014)." (Line 447-449)

Section 4.1 I don't see any discussion of how the different treatments influenced water use.

R: In the section 4.1, we only focused on the comparison of tree transpiration and the possible mechanisms between the two tree species under control condition (treatment of BC), and the influence of changed precipitation patterns on water use of coexisting trees was emphasized in section 4.2: "In this study, only S. superba experienced significant increase of WUEi under DD treatment, indicating its better ability to cope with intensified drought stress. Differing from S. superba, M. macclurei did not significantly change WUEi under DD and ED treatments. This less adjustment of WUEi together with its higher transpiration under relatively drier seasons implied an disadvantage for M. macclurei when facing water stress... In our study, compared with the BC condition, the DD and ED treatments did not significantly change the Huber values (p > 0. 05), but did pose an obvious influence on the utilization of water from distinct water sources, especially for S. superba. Though the relatively higher use of rainfall water has decreased the use of groundwater under DD treatment, the prolonged dry condition still promoted S. superba to utilize deeper soil water (40-60 cm)." (Line 472-486)

Table 2. The letters used to discern differences between BC, DD, and ED treatments are quite confusing. For example, why are different letters used for Branch As:Al (b, c, d) compared to WUEi (a, b). This almost implies that WUEi and As:Al were compared, when they clearly were not.

R: To emphasize the effects of tree species and precipitation treatment separately, we have redone the statistical analysis and the results were listed in Table 3 (new version of the manuscript). Corresponding modification of the results was described in the text. (Line 363-371)

Figure 2. The letters used to show differences in SWC are too complicated. Please remove and just report the statistical findings. Why are lower case and capital letters both used? The Figure legend does not explain any of this.
R: Different small letters indicate differences among the three treatments within the same tree species. Different capital letters indicate differences between tree species for a single treatment. The letters were removed in Figure 2 and the statistical findings were reported as suggested: "According to the statistical analysis, the DD treatment possessed significantly lower SWC values for majority of the experimental months, with approximately 5%-30% decline compared to BC and ED treatments (p < 0.05), and no difference was observed between the BC and ED treatments in the wet season (p > 0.05)." (Line 335-338)

Figure 3. Instead of splitting the daily E into dry, spring, and summer, I would plot daily E along one time axis. The way this figure is currently set up, the time intervals are different between panels a, c, e, and b, d, f. A better way is to highlight the different periods of the precipitation in one panel, and have M. Macclurei on top, and S. superba on the bottom. Also, why is ED lower during the dry season (Oct-Mar) if precipitation was not excluded during this time?

R: The Figure 3 was re-plotted accordingly by following reviewer's suggestion .

Figure 4. No statistics here. Why?

R: We have displayed the statistical analysis in Figure 4. The corresponding results of statistical analysis are also described in the manuscript. (Line 379-392)

Please also note the supplement to this comment:
https://www.biogeosciences-discuss.net/bg-2019-392/bg-2019-392-AC1-supplement.pdf

---

## Author Comment (AC2) · 7 Jan 2020

Reviewer 1 General comments: In this manuscript, Ouyang et al. present the results of a precipitation manipulation experiment in which the dry season was exacerbated or lengthened (along with a compensating increase in wet season water supply). The authors report a number of traits for the two dominant tree species at their experimental site, along with species-specific transpiration and water-use patterns. They conclude that the two dominant species show contrasting water-use strategies and their findings have important implications for the survival of these species under a changing climate

regime. The experimental setup and amount of data collected is impressive, but the manuscript has some key issues that I believe preclude publication at this time. Below, I have listed some broader feedback about the manuscript, followed by smaller line by line comments. 1)In general, I think the most novel aspect of this manuscript is the water uptake depth data. However, I think there is significant methodological detail missing about how water uptake depth was partitioned and there seems to be no statistical analysis comparing these data across treatments or species. I also think it should be mentioned that water samples were collected during a very limited time frame and that plant water use could have shifted during the experiment, as could precipitation 18O. Further, the authors make claims about these data (primarily differences between species), that look unfounded to me but could potentially be shown statistically. In sum, I don't think the water uptake depth analyses are valid as is. Without the water uptake depth data, I think the manuscript is fairly simple, merely showing the effects of the precipitation manipulation on transpiration, along with a few traits. 2)Related to the above point, I think there are some large gaps in the methods that make a solid interpretation of the results difficult. In particular, I think the manuscript is missing detail on leaf 13C sampling, lacks a discussion on the age of the bulk material they sampled for 13C analysis and what this means for their interpretations, neglects description of their water uptake depth model, and lacks some additional smaller details on sampling that I've highlighted below.

Response (R): We sincerely appreciate that the reviewer pointed out the pertinent problems of our manuscript and detailed modification suggestions. We have thoroughly checked and revised the manuscript by following reviewer's suggestions, and respond as well as explain item-by-item the questions as follows. The methodological detail about how water uptake depth was partitioned was added in the manuscript: "Four rainfall samples from four precipitation events were collected for the isotope analysis. Previous root sampling analysis has verified that more than 80% of the total root biomass for S. superba and 80% of the absorbing roots of M. macclurei were distributed in the surface soil layer (0-60 cm) (Hao and Peng 2009; Li 1984). Therefore,

soil samples around the sampled trees were collected from the upper soil layers (0-20 cm), middle soil layers (20-40cm) and deeper soil layers (40-60 cm) with soil cores from each experimental plot. Water from a small well near the experimental plots was collected as the groundwater and kept in the laboratory at 0-5 °C for further analyses (Sun et al., 2018). " (Line 287-296)

In addition, according to reviewer's suggestion, we recalculate the proportions of water resource use based on the measured isotope data and display the statistical analysis in Figure 4. The corresponding results of statistical analysis are also described in the manuscript: " According to the statistical results, the utilization of rainwater and soil water of M. macclurei trees showed no significant treatment-difference, while the DD and ED treatments significantly decreased its utilization of groundwater (Figure 4). However, the changed precipitation pattern posed a significant influence on the water use proportions of S. superba from different water resources ($p < 0.05$)....Furthermore, the two dominant tree species shared similar water use proportion under the control condition, and M. macclurei used more soil water (0-60cm) than the S. superba under DD treatment. Comparatively, M. macclurei utilized more rainwater, while S. superba was inclined to make use of more groundwater under ED treatment ($p < 0.05$)." (Line 379-392)

The detailed information about leaf 13C sampling and analysis was also added in the manuscript: "In this study, the obtained fresh mature leaves (the pre-treated details was described in section 2.6) were oven dried (80âŮęC, 48 h), crushed to a powder and sieved through a 100 mesh. The samples passing through the mesh were oxidized with an elemental analyzer (VARIO EL3, Elementar, Germany) and analyzed for $\delta$13C by mass spectrometry (DELTA V Advantage, Thermo Scientific, USA) using Pee Dee Belemnite (PDB) limestone as the standards. The $\delta$13C value (‰ was calculated from the following equation..." (Line 260-266)

3)In many places throughout the manuscript, the authors claim their results speak to the competitive ability of these species using words like "competition", "success", "coexistence", "tolerance", etc. I do not think the data support these claims since the manuscript merely shows patterns in water use between the two species and some additional traits. I think this language should be toned down and I've highlighted areas where this issue came up.

R: The terminology words such competition, success or tolerance were deleted or revised to avoid high-toned. We think the use of the terminology "coexistence" might be reasonably, because the two dominant species in this area, M. macclurei and S. superba shared and competed for the resources with different distribution of root biomass, various water use and growth indexes, thus, their coexistence could be a point in this study.

4)Certainly, this is not a big issue at this stage, but I think the manuscript could use some significant editing for grammar and sentence structure.

R: We have read and done our best to make revisions throughout the text to avoid grammar mistakes and long sentences.

Specific comments: L1-2: I would really recommend altering the title. I do not really think the study pertained to "acclimation traits", nor were any traits "reshaped". Further, I think that saying they "altered their coexisting relation" is not supported by the data (my point #3 above).

R: The title was changed to "Species-specific transpiration and water-use patterns of two dominant coexisting tree species under manipulated rainfall in a low-subtropical secondary evergreen forest"

L27: See #3 above about claims of competitiveness.

R: This sentence was changed to "...that M. macclurei has more survival and growth advantages in this subtropical forest." (Line 27-28)

L29-30: How are you defining drought tolerance? It seems like the two species had fairly similar responses to the precipitation manipulation in terms of transpiration and

there was no evidence that either species was water stressed (point 3# above).

R: Actually, due to the abundant rainfall, drought events do not happen very often in south China, but previous reports also claimed many tropical areas with rich species have already experienced little or no rain falls during dry seasons and upper soil layers might undergo severe drying (Goldstein et al., 2008; Liu et al., 2010; Gao et al., 2015) (Line 88-91). In this study, S. superba allocated much less root biomass on surface soil than M. macclurei (47% vs. 72%), and as shown in the statistical results, the S. superba was inclined to use more groundwater than M. macclurei under ED treatment. Therefore, we think S. superba might be drought tolerant under potential drought in the future. To make it less controversial, this sentence was changed to "Therefore, under the seasonal drought caused by uneven distribution of rainfall in the future, M. macclurei that inclines to use shallow soil water would adopt a drought-avoidance strategy, whereas S. superba being able to uptake deeper soil water would be drought tolerant. (Line 29-31)"

L31: See#3 above about the coexistence terminology.

R: As we had explained in the previous response, M. macclurei and S. superba shared and competed for the resources, and had different distribution of root biomass, various water use and growth indexes, thus, the coexistence terminology might be reasonable.

L78-80: I do not think it is well supported in the literature that isohydric species tend to occur in mesic areas. They can and do exist most everywhere.

R: This sentence was changed to "Isohydric species, however, are often regarded as drought avoiders as they can avoid drought-induced hydraulic failure by way of strict stomatal control and relatively constant minimum leaf water potential (McDowell et al., 2008)" (Line 75-78)

L128-135: It seems like #1 and #2 are really similar in the sense that they both describe traits and water-use patterns. Maybe the authors could separate these objectives out

into one about tree water use (and water uptake depth), and one about how traits mediate tree water use patterns.

R: Thanks for the suggestion, we have rewritten the objectives: "Therefore, main objectives of this study are 1) to investigate the effects of manipulated precipitation conditions on spatial-temporal water use patterns of S. superba and M. macclurei in this subtropical forest; 2) to understand the potential mechanism for the varied responses of tree transpiration to the changed precipitation patterns by examining the variations in morphological adjustment, such as Huber values (As: Al), the intrinsic water use efficiency, and the contributions of water resources to the tree transpiration." (Line 135-141)

L161-164: I think it would be very helpful for the reader if there was a simple diagram (or perhaps labels added on to a time series figure) showing when precipitation was excluded or added back in for each treatment.

R: To describe the manipulated precipitation treatments more clearly, we listed the detailed information of excluding and irrigating water under DD and ED treatment in Table 1. "The exclusion of precipitation was achieved automatically by a tarpaulin covering approximately 67% of the area of the DD and ED plots. To guarantee the equal total annual rainfall, approximately equivalent amounts of excluded water were pumped into these plots several times (4-8 times) during wet seasons (from June to September for DD and ED treatments) (Table 1)." (Line 174-178)

L178: In order to interpret the effects of the precipitation manipulation, there needs to be information about how the experimental year's climate related to average site climate. For example, if this was a really wet year, there may be no reason to expect significant treatment effects in the first place.

R: We have described the long term climate of the study site in section of "Site description", and we also presented the climatic parameters especially the precipitation of the experimental year in section 3.1: "Total precipitation at the research site during the

experimental period was 2094 mm. The precipitation was unevenly distributed and occurred mainly between April and September, accounting for approximately 84% of the annual total. It was noticeable that the heaviest precipitation with a value of 498.6 mm occurred in August, while the lightest precipitation occurred in February with only 2.7 mm. (Line 327-332)" Data has obviously shown the unevenly distributed precipitation, which could make our experimental design sense.

Yes, it was really a wet year, as the total precipitation being of 2094 mm during the experimental period, but the rainfall manipulation that excluded 67% of precipitation under DD and ED treatment (the amounts of excluded water were shown in Table 1) still had a non-negligible effect on tree transpiration, especially during the periods of dry and spring drought season (Figure 3).

L216: What depth were these samples collected at?

R: "soil samples (0-30 cm) were periodically collected in the experimental plots to measure the soil water contents (SWC) by gravimetric method." (Line 222)

L252-290: I think some text should be added (either in the methods or discussion) that clarifies what the 13C signal would represent. If these leaves had been around prior to the experiment, their bulk tissue 13C would incorporate the 13C signal from climatic conditions at the time of leaf expansion, the carbon used to make those leaves, and any dynamics influencing non-structural carbohydrates since leaf expansion. Any information on the life span of these leaves would help here, or simply a caveat that the 13C signal could be complicated.

R: The detailed information about leaf 13C sampling and analysis was also added in the manuscript: "In this study, the obtained fresh canopy leaves (the pre-treated details was described in section 2.6) were oven dried (80âŲẹC, 48 h), crushed to a powder and sieved through a 100 mesh. The samples passing through the mesh were oxidized with an elemental analyzer (VARIO EL3, Elementar, Germany) and analyzed for $\delta$13C by mass spectrometry (DELTA V Advantage, Thermo Scientific, USA) using Pee Dee

Belemnite (PDB) limestone as the standards. The $\delta$13C value (‰ was calculated from the following equation..." (Line 262-268)

Since the S. superba and M. macclurei are evergreen tree species with perennial leaves, 13C value of mature leaf is generally considered to be constant after being fixed in the leaves, and thus, the use of mature leaves for 13C sampling and analysis at the end of the experiment are reasonable.

L275: When were rainfall samples collected? Were these multiple samplings or were there 4 replicates of one rainfall event? If it is the latter, I don't think you can assume that this rainfall event is representative of all rainfall.

R: The collected rainfall sample were taken from four precipitation events. "Four rainfall samples from four precipitation events were collected for the isotope analysis." (Line 287-288)

L286: Please describe in detail what IsoSource is and the methodology behind how it partitions water uptake depth.

R: The IsoSource, a mixing model software, is developed and introduced by Phillips and Gregg (2003). It is designed for situations in which n isotopes are being used and more than n+1 sources are likely to contribute to a mixture. IsoSource uses stable isotope data to calculate feasible ranges of source contributions. Detailed information about this software was presented in Phillips and Gregg (2003). In this study, the xylem water was regarded as the mixture, and different water samples for isotope analysis included the rain, soil water from different soil layers, and groundwater. First, all possible combinations of source proportions that sum to 100% are calculated in user-specified increments (2% in our study). Second, the predicted isotope values of the mixture are computed using linear mixing model equations that preserve mass balance (Phillips 2001). Isotope values of computed mixtures are then compared with the observed isotope values; the range of combinations that match within a user-specified tolerance value (0.05% in our study) is then described.

L343-350: What are these percentage reductions in comparison to (i.e., what counted as dry versus wet seasons)?

R: These percentage reductions are in comparison to wet seasons. In the new version of manuscript, we deleted this part to avoid the unnecessary description.

L364-378: As mentioned in point #1, this sections needs some statistical analysis to be able to draw any conclusions from the data.

R: The results of statistical analysis are also described in the manuscript: "According to the statistical results, the utilization of rainwater and soil water of M. macclurei trees showed no significant treatment-difference, while the DD and ED treatments significantly decreased its utilization of groundwater (Figure 4). However, the changed precipitation pattern posed a significant influence on the water use proportions of S. superba from different water resources ($p < 0.05$)....Furthermore, the two dominant tree species shared similar water use proportion under the control condition, and M. macclurei used more soil water (0-60cm) than the S. superba under DD treatment. Comparatively, M. macclurei utilized more rainwater, while S. superba was inclined to make use of more groundwater under ED treatment ($p < 0.05$)." (Line 379-392)

L390-391: Is this saturation model warranted for the data considering that most of the responses seem to be fairly linear? Is there a first principles reason to expect a saturation relationship?

R: Generally, the response of tree transpiration to PAR is generally linear when the PAR values are relatively low, while non-linear (saturation) relationships would be observed as PAR further increase. Considering the wide range of PAR in our study (0-50 mol m-2 d-1), we thus used the exponential saturation model to explore the response of tree transpiration to PAR.

L424: I'm confused as to how Huber value can be used to understand how much water a species has access to.

R: This sentence was revised as "Results indicated that S. superba had significantly larger Huber value (As:Al) (Table 3), possibly meaning a less investment on leaf biomass but a better efficient transport system (Zhu et al., 2014)." (Line 447-449)

L427: See point #3 above regarding the "drought-tolerant"terminology.

R: As we have explained above, to make this terminology less controversial, we revise this sentence as "This character could lead to restraining of transpiration and better transport efficiency, and thus to drought-tolerant for S. superba when severe drought occurs. (Line 451-453)"

L433-434: I'm not sure this is supported by the data (especially since there are no stats). The bars seem to be similar in size and the error bars seem to overlap.

R: The results of statistical analysis are added in the revised manuscript and Figure 4. "According to the statistical results, the utilization of rainwater and soil water of M. macclurei trees showed no significant treatment-difference, while the DD and ED treatments significantly decreased its utilization of groundwater (Figure 4). However, the changed precipitation pattern posed a significant influence on the water use proportions of S. superba from different water resources (p < 0.05)....Furthermore, the two dominant tree species shared similar water use proportion under the control condition, and M. macclurei used more soil water (0-60cm) than the S. superba under DD treatment. Comparatively, M. macclurei utilized more rainwater, while S. superba was inclined to make use of more groundwater under ED treatment (p < 0.05)." (Line 379-392)

L489-490: I'm not sure this claim is supported since the treatment effects seemed to be fairly similar in the two species. Perhaps specify or cut this sentence?

R: This sentence was deleted as we actually don't have the experimental data to support that.

All figure and tables: Please specify what the +/- means in the tables (standard error,

deviation?), what the lettering notation indicates, and what error bars represent.

R: Done. The +/- means mean values ± standard deviation. Different small letters indicate differences among the three treatments within the same tree species (p <0.05); Different capital letters indicate differences between tree species for a single treatment (p <0.05).

Fig. 1: During what hours were these daily values calculated? It might be more relevant to present mean daytime PAR and VPD.

R: Yes, we also think that the mean daytime values of PAR and VPD would be more relevant, however, the values were measured and offered by the Heshan National Ecological Station and only daily values presented.

Fig. 3: I think the clarity of this figure could be improved. In general, it is hard to parse out trends due to the experiment since the points are so close together. It is also hard to interpret the panels for each species since they encompass different and overlapping time points. Perhaps clarify what manipulation was occurring in each panel, or maybe make one longer time series graph with all the treatment times labeled?

R: The Figure 3 was re-plotted according to the suggestion. We used three separate graphs to describe tree transpiration of two species during the whole experimental period under BC, DD and ED treatment, respectively.

In terms of the treatment times, "The exclusion of precipitation was achieved automatically by a tarpaulin covering approximately 67% of the area of the DD and ED plots. To guarantee the equal total annual rainfall, approximately equivalent amounts of excluded water were pumped into these plots several times (4-8 times) during wet seasons (from June to September for DD and ED treatments) (Table 1)." (Line 174-178)

Please also note the supplement to this comment:
https://www.biogeosciences-discuss.net/bg-2019-392/bg-2019-392-AC2-

supplement.pdf